# Understanding public trust in information about interim nuclear waste storage: The roles of acceptance, gender, and proximity

Aizhan Zabirova[1], Hitomi Matsunaga[1]*, Yuya Kashiwazaki[1], Xu Xiao[1], Thierry Schneider[2], Noboru Takamura[1]

1 Department of Global Health, Medicine and Welfare, Atomic Bomb Disease Institute, Nagasaki University Graduate School of Biomedical Sciences, Nagasaki, Japan, 2 Nuclear Protection Evaluation Centre (CEPN), Fontenay-aux-Roses, France

* hmatsu@nagasaki-u.ac.jp

## Abstract

Following the accident at the Fukushima Daiichi Nuclear Power Station in March 2011, the Japanese government implemented extensive decontamination and waste management measures, including the establishment of an interim storage facility in Okuma and Futaba. This study aims to analyze residents' trust in the information provided by the public authorities regarding this facility and to identify factors influencing that trust. A survey was conducted among 1,558 former residents of Tomioka, Okuma, and Futaba who were 18 years or older and held resident cards for one of these towns in both March 2011 and 2024. The questionnaire collected data on demographic variables; intention to return; thoughts on the storage facility; trust in public information; concerns about radiation-related genetic risks and negative perceptions due to the nuclear accident; and mental health status, assessed using the mental component of the SF-8 health survey. The results showed that 57.7% of respondents trusted the information provided by the public authorities regarding the storage facility. Factors significantly associated with trust included their acceptance of the facility, lack of concerns about radiation-related genetic risks, lack of concerns about negative images resulting from the nuclear accident, and good mental health status. A logistic regression analysis indicated that acceptance of the facility and good mental health significantly increased the likelihood of trusting information provided by the public authorities, while concerns about genetic risks and negative images significantly decreased it. These findings underscore the vital importance of recognizing and addressing residents' concerns about health risks and negative perceptions related to nuclear waste management. By supporting the mental well-being of the community and fostering transparent communication, authorities can build trust and improve the acceptance of waste management facilities. Actively engaging with residents and responding to their worries facilitates the identification of adequate and acceptable recovery efforts after a nuclear accident.

**Data availability statement:** All relevant data are included in the paper and its Supporting Information files.

**Funding:** This work was supported by the Research Project on the Health Effects of Radiation organized by the Ministry of Environment, Japan through grants awarded to Noboru Takamura.

**Competing interests:** The authors have declared that no competing interests exist.

## Introduction

Following the accident at the Fukushima Daiichi Nuclear Power Station (FDNPS) on March 11, 2011, the Japanese government and responsible authorities embarked on a large-scale decontamination project [1]. Under the Act on Special Measures, the national government designated Special Decontamination Areas in 11 municipalities of Fukushima Prefecture and Intensive Contamination Survey Areas in 104 towns across eight prefectures and directly conducted soil decontamination, which was completed by the end of 2017 in all Special decontamination areas. An Interim Storage Facility (ISF) was constructed after getting the agreement from Fukushima Prefecture in 2014 to ensure the safe management and storage of contaminated soil and wastes, followed by approvals from Okuma and Futaba Towns in 2015. The entire site covers about 16 km$^2$. The Ministry of the Environment issued a "Basic Concept" in 2016 stipulating strict radiological safety criteria and a maximum cesium-137 concentration of 8000 Bq/kg for soil reuse. The ISF is designed to handle and store materials securely and efficiently for up to 30 years, during which efforts will focus on volume reduction, recycling, and determining final disposal methods. While a portion of the stored materials is intended for final disposal outside the prefecture, another portion is being evaluated for recycling and reuse in applications like public works. These plans remain under development, subject to ongoing technological advancements and social acceptance [2]. The facility includes areas for receiving and separating materials, storing soil, and storing waste. This classification and continuous monitoring of dose rates around the facility helps to prevent unfounded fears and builds confidence in its controlled safety [3]. One of the most complex and multilayered areas of this process has involved working with the people affected by the disaster, including providing them compensation for damages, mental and physical health support, and long-term assistance. These people are mainly residents of the Fukushima Prefecture, but the group includes all those affected by the radiation accident [4]. Radiation safety information centers, support and communication centers, and centers for measuring radiation levels in food have been built in the region [5].

The process of selecting sites for radioactive waste storage and disposal invariably affects the public interest and is accompanied by significant challenges related to the perception of radiation risks and their involvement in the decision-making process. There are concepts such as NIMBY ("Not In My Backyard"), LULU ("Local Undesirable Land Use"), BANANA ("Build Absolutely Nothing Anywhere Near Anything"), and NOTE ("Not Over There Either") that reflect the public's reluctance to accept temporary storage or permanent disposal of radioactive waste close to where they live. These terms describe residents' objections to these types of facilities [6]. The main issue underlying such objections is the inequitable distribution of risks and burdens [7]. Situations where, as in the case of the Fukushima accident, radioactive waste is generated as a result of unforeseen events, rather than a planned process, present a particular challenge. In contrast to planned regular disposal procedures, accidents create an additional element of uncertainty that increases public concern and emphasizes the importance of trust in the public authorities and the information they provide on the temporary storage of radioactive waste. This uncertainty is reflected

in the difficulty of predicting future scenarios, risk levels, and outcomes associated with managing and storing radioactive waste generated by accidents. In contrast to planned disposal procedures—usually guided by well established regulatory frameworks and rigorous risk assessments—accidents introduce conditions and variables that are less thoroughly understood. As a result, it becomes harder to ascertain how, where, and under what conditions such waste will be stored, as well as the long-term environmental and health implications. This reduced certainty makes it challenging for the public to trust safety measures or to have confidence in the adequacy and fairness of risk distribution. As previous studies have shown, in the aftermath of major accidents like Chernobyl and Fukushima, public distrust in governmental actions tends to increase [8]. Distrust in public authorities following nuclear accidents has been shown to negatively affect mental health, including increased anxiety and altered perceptions of health risks. Previous studies examining the Fukushima and Three Mile Island accidents identified clear associations between distrust in governmental institutions and mental health outcomes [9]. Additionally, risk perception—captured through residents' concerns about genetic consequences shapes trust levels. Therefore, this study analyzes the levels of residents' trust in information provided by public authorities regarding the ISF in Okuma and Futaba, incorporating mental health and risk perception as critical influencing factors. The accident also significantly damaged the reputations of these residents' towns, creating persistent negative images associated with nuclear contamination and risk. As Gusterson (2011) noted, the accident caused profound cracks in the "social containment vessels" surrounding nuclear energy, shaking previously unquestioned assurances that nuclear reactors were safe neighbors. These negative perceptions deeply affected local residents' trust in information provided by the public authorities [10]. As their towns became closely identified with the nuclear disaster, residents faced increased skepticism, suspicion, and uncertainty regarding official assurances about safety and nuclear-related issues. This possibly included the information they received about the ISF. Residents' concerns about the negative image of their towns resulting from the nuclear accident are directly relevant to understanding variations in their trust toward public authorities in that question.

Understanding this influence will shed light on the specificity of the level of trust in post-accident information in the context of nuclear waste management among residents within a 10–20 km radius of the FDNPS.

## Materials and methods

### Participants

The study was conducted in the towns of Tomioka, Okuma, and Futaba, in Japan's Fukushima Prefecture..

The subjects of this study were current and former residents of the towns mentioned above who possessed resident cards as of March 2011 and who still retained them as of 2024. Only residents aged 18 years and older in March 2011 were included in the study. We included a total of 1558 responses (536 from Tomioka, 580 from Okuma, and 442 from Futaba) in the analysis. Written consent was obtained from all participants, and the study was approved by the Biomedical Sciences Ethics Committee of Nagasaki University, protocol number 23081805. Permission was also obtained from the municipalities of the studied towns.

### Questionnaire

The questionnaire used in this study was developed based on a survey used in previous studies conducted in the Fukushima Prefecture [11, 12, 13]. The survey aimed to determine participants' trust in the information provided by public authorities regarding the ISF. Data were collected on demographic variables, including town of residence, age, sex, and current place of residence. Participants were also asked about their intention to return to their hometowns, with response options of "already returned," "want to return," "undecided," and "do not want to return." The questionnaire included three questions related to removed soil and the ISF. Participants were first asked whether they wanted to know more about the removed soil, with those responding "yes" given multiple choices for the specific topics they wanted to know more about. Then, they were asked about their thoughts on having an ISF in Okuma and Futaba, with options of "accept," "not

sure," or "do not accept." Finally, participants were asked to rate their trust in the information provided by public authorities regarding the ISF using a 4-point scale (1 = yes, 2 = probably yes, 3 = probably no, 4 = no). The survey also included two questions about participants' concerns: one regarding radiation-related genetic risks and another about the negative images to their town resulting from the nuclear accident, both of which were answered using the same 4-point scale as in the trust in information question describe above. To determine respondents' mental health status, we used the Health-related Quality of Life (HR-QOL) Short Form-8 (SF-8) that originally measured eight dimensions: general health, physical function, physical concerns, bodily pain, vitality, social functioning, mental health, and emotional concerns. The first four scales show a Physical Components Summary and the second a Mental Components Summary (MCS). In our questionnaire, we only used MCS questions. The mean value for the Japanese population was $50 \pm 10$, which was used to dichotomize the HR-QOL scores in this study [14].

## Statistical methods

Incomplete responses were considered missing by the user, and we performed pairwise exclusions for missing values. We also excluded answers from those who "preferred not to reveal" their sex, as the number of such responses was low. "Age" was thus recalculated as younger (< 64 years) and older (≥ 65 years) residents. "Trust" and "Concern about radiation-related genetic effects; negative image due to nuclear accident" were converted from a 4-point Likert-type scale to "yes" or "no" responses. Factors associated with trust in the information provided by public authorities regarding the ISF were identified using chi-square tests. Factors independently distinguishing between groups were identified using logistic regression analyses. Two separate logistic regression models were used to examine how two different types of risk concerns, genetic risks associated with radiation (Model I) and negative image due to a nuclear accident (Model II), affect participants' trust in the information provided by public authorities regarding the ISF. Variables with three or four categories were divided into two, to make binominal logistic regression possible. Thus, we grouped "inside Hamadori" and "outside," which includes the Nakadori and Aizu regions, and the "Outside Fukushima" group. Thoughts on having an ISF in Okuma and Futaba were divided into the groups "accept" and "other," which included "not sure" and "do not accept." The data were statistically analyzed using SPSS software (Statistical Package for Social Science) Statistics 29.0 (IBM Armonk, New York, USA). A $p$-value of less than 0.05 was considered statistically significant.

## Results

### Participants' characteristics

The number of participants from Tomioka was 536 (34.4%), while 580 (37.2%) were from Okuma and 442 (28.4%) were from Futaba. A majority of all participants were older than 65 years (64.5%). More than half (52.1%) now live in the Hamadori region, with fewer residing outside the Fukushima prefecture (26.3%) or in the Nakadori and Aizu regions (21.5%). Among the residents, there were almost equal numbers of males (55.5%) and females (43.7%) with 0.8% choosing not to reveal their sex. Most participants did not want to return to the region (57.0%), and about 24.7% remain undecided. A minority have already returned (6.9%) or want to return (11.4%). All three towns' residents showed a positive outlook toward the ISF in Okuma and Futaba (65.4%). Moreover, many participants showed trust in information provided by public authorities regarding the ISF (57.7%); among them, roughly 15% replied "yes" and around 43% said "probably yes" when asked if they trust it. The number of those who "probably do not" trust the information given to them was 478 (31.1%), while and those who chose "no" (11.2%) were the smallest group. Concerns about radiation-related genetic risks were relatively low, at just 37.0%: 11.8% responded as "yes," 25.2% answered "probably yes." These participants (almost 69% of them) also tended to have concerns about the negative image to their town resulting from the nuclear accident, with 29% answering "yes" and around 40% saying "probably yes" when asked about this. Most reported good mental health (57.2%) (Table 1). Around 75% of participants want to know more about the reuse of removed soil. For those who indicated that they were interested in knowing more, we offered specific

**Table 1. Participants' characteristics.**

| Variable | Reference | N | % |
|---|---|---|---|
| Town (n = 1558) | Tomioka | 536 | 34.4 |
| | Okuma | 580 | 37.2 |
| | Futaba | 442 | 28.4 |
| Age (n = 1531) | ≤64 | 543 | 35.5 |
| | ≥65 | 988 | 64.5 |
| Sex (n = 1551) | Male | 861 | 55.5 |
| | Female | 678 | 43.7 |
| | Prefer not to say | 12 | 0.8 |
| Current residence (n = 1523) | Hamadori region | 794 | 52.1 |
| | Nakadori and Aizu regions | 328 | 21.5 |
| | Outside Fukushima Prefecture | 401 | 26.3 |
| ITR (n = 1543) | Already returned | 107 | 6.9 |
| | Want to return | 176 | 11.4 |
| | Undecided | 381 | 24.7 |
| | Do not want to return | 879 | 57.0 |
| Want to know more about removed soil (n = 1534) | Yes | 1144 | 74.6 |
| | No | 390 | 25.4 |
| Thoughts on having an ISF in Okuma and Futaba (n = 1527) | Accept | 999 | 65.4 |
| | Unsure | 353 | 23.1 |
| | Do not accept | 175 | 11.5 |
| Trust the information provided by public authorities regarding the ISF (n = 1536) | Yes | 229 | 14.9 |
| | Probably yes | 657 | 42.8 |
| | Probably no | 478 | 31.1 |
| | No | 172 | 11.2 |
| Concerned about radiation-related genetic risks (n = 1522) | Yes | 180 | 11.8 |
| | Probably yes | 383 | 25.2 |
| | Probably no | 554 | 36.4 |
| | No | 405 | 26.6 |
| Concerned about a negative image resulting from the nuclear accident (n = 1544) | Yes | 448 | 29.0 |
| | Probably yes | 613 | 39.7 |
| | Probably no | 347 | 22.5 |
| | No | 136 | 8.8 |
| Mental health (HR-QOL SF8) (n = 1543) | <50 | 660 | 42.8 |
| | ≥50 | 883 | 57.2 |

*Note.* ITR - intention to return; ISF - interim storage facility, HR-QOL SF8; Health-Related Quality of Life Short Form 8.

options such as: "how to reuse it," "health effects," "environmental impact," and "reuse and other concerns." It was possible for them to indicate multiple answers to this question. Our results showed that, among those who trusted the information given to them, 64.9% wanted to know how the soil would be reused, followed by 49.9% who were interested in the environmental effects, 42.6% wanted to know more about health effects, and 3.9% choose reuse of removed soil and other issues. Among those who indicated that they mostly do not trust the information, the greatest area of interest was in the soil's environmental impact (61.3%), followed by health effects (59.5%), how the soil will be reused (48.0%), and recycling and other issues (7.1%) (Fig 1).

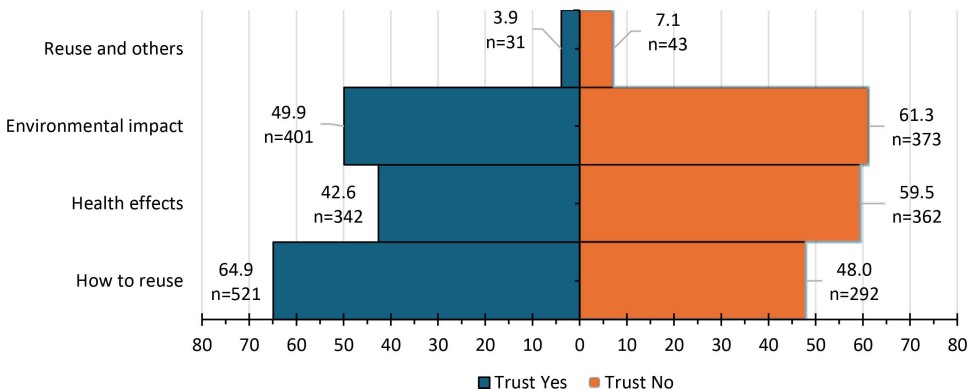

**Fig 1. Specific information residents want to know more about ISF.**

### Factors associated with trust in the information provided by public authorities regarding the ISF

The distribution of trust varies significantly between sexes ($p = 0.005$). Males (59.2%) exhibit higher trust levels, compared to females (40.4%). Trust levels vary significantly based on current residence ($p = 0.011$). Those currently residing in the Hamadori area (53.8%) have higher trust levels, compared to Nakadori and Aizu area residents (22.9%) and those living outside Fukushima (23.3%). Acceptance of having an ISF in Okuma and Futaba is not surprisingly strongly associated with trust levels ($p < 0.001$). Those who accept the ISF (79.4%) show markedly higher trust, compared to those unsure (18.1%) and those who do not accept it (2.5%). Trust levels are significantly affected by concerns about radiation-related genetic risks ($p < 0.001$). Individuals not concerned about these risks (79.7%) exhibit much higher trust levels, compared to those who are concerned (20.3%). Individuals not concerned about the negative image from the nuclear accident (37.7%) have higher trust levels, compared to those who are concerned (62.3%), ($p < 0.001$). Mental health status, as measured by the HR-QOL SF8, shows a significant association with trust levels. Individuals with good mental health (67.2%) display higher trust, compared to those with poor mental health (32.8%, $p < 0.001$) (Table 2).

### Logistic regression analysis

Model I shows that those who accept having an ISF in Okuma and Futaba are significantly more likely to trust the information provided to them by the authorities (OR: 3.91, CI: 3.02–5.05, $p < 0.001$). Those concerned about genetic risks are significantly less likely to trust this information (OR: 0.19, CI: 0.15–0.25, $p < 0.001$). Those with good mental health are significantly more likely to trust the information (OR: 1.75, CI: 1.36–2.24, $p < 0.001$). In Model II, males are significantly more likely to trust the information, compared to females (OR: 1.31, CI: 1.04–1.65, $p < 0.05$). Accepting the ISF is, again, significantly associated with greater trust (OR: 4.08, CI: 3.20–5.19, $p < 0.001$). Those concerned about a negative image are significantly less likely to trust the information (OR: 0.56, CI: 0.43–0.72, $p < 0.001$). Good mental health is, again, significantly associated with higher levels of trust (OR: 2.01, CI: 1.59–2.53, $p < 0.001$) (Table 3).

Males show slightly higher trust in both models, but significantly so in Model II. Residence inside or outside Fukushima does not significantly affect trust in either model. Accepting the ISF in Okuma and Futaba significantly increases trust in both models. Concerns about genetic risks and negative images significantly decrease trust in both models. Good mental health significantly increases trust in both models.

**Table 2. Factors associated with trust in information provided by public authorities.**

| Variable | Reference | Trust 57.7% (886) | Do not trust 42.3% (650) | p-value |
|---|---|---|---|---|
| Age | <64 | 35.7 (312) | 36.0 (229) | 0.913 |
| | ≥65 | 64.3 (563) | 64.0 (407) | |
| Sex | Male | 59.5 (522) | 52.3 (334) | 0.005 |
| | Female | 40.5 (356) | 47.7 (305) | |
| Current residence | Hamadori area | 53.8 (467) | 49.8 (316) | 0.011 |
| | Nakadori and Aizu area | 22.9 (199) | 20.2 (128) | |
| | Outside Fukushima | 23.3 (202) | 30.1 (191) | |
| ITR | Already returned | 8.1 (71) | 5.6 (36) | 0.101 |
| | Want to return | 12.3 (108) | 10.2 (66) | |
| | Undecided | 23.5 (206) | 26.3 (170) | |
| | Do not want to return | 56.2 (493) | 58.0 (375) | |
| Want to know more about removed soil | Yes | 73.1 (640) | 76.2 (487) | 0.171 |
| | No | 26.9 (236) | 23.8 (152) | |
| Thoughts on having an ISF in Okuma and Futaba | Accept | 79.4 (693) | 46.8 (300) | <0.001 |
| | Unsure | 18.1 (158) | 29.3 (188) | |
| | Do not accept | 2.5 (22) | 23.9 (153) | |
| Concerned about radiation-related genetic risks | Yes | 20.3 (178) | 60.2 (379) | <0.001 |
| | No | 79.7 (700) | 39.8 (251) | |
| Concerned about a negative image resulting from the nuclear accident | Yes | 62.3 (549) | 77.3 (498) | <0.001 |
| | No | 37.7 (332) | 22.7 (146) | |
| Mental health (HR-QOL SF8) | Good | 67.2 (592) | 43.8 (282) 56.2 | <0.001 |
| | Poor | 32.8 (289) | (362) | |

*Note.* chi-square tests, ITR; intention to return, ISF; interim storage facility, HR-QOL SF8; Health-Related Quality of Life Short Form 8

**Table 3. Logistic regression analyses for trust in information provided by public authorities.**

| Variable | Model I Genetic risk concerns OR (95%CI) | Model II Negative image concerns OR (95%CI) |
|---|---|---|
| Gender (male/female) | 1.17 (0.87–1.57) | 1.31 (1.04–1.65) * |
| Residence (inside/outside Hamadori) | 0.97 (0.72–1.31) | 1.05 (0.79–1.39) |
| Thoughts on having an ISF in Okuma and Futaba (accept/other) | 3.91 (3.02–5.05) ** | 4.08 (3.20–5.19) ** |
| Concerned about radiation-related genetic risks (yes/no) | 0.19 (0.15–0.25) ** | |
| Concern about a negative image resulting from the nuclear accident (yes/no) | | 0.56 (0.43–0.72) ** |
| Mental health (good/poor) | 1.75 (1.36–2.24) ** | 2.01 (1.59–2.53) ** |

*Note.* Logistic regression analyses, Trust (Yes) – the reference group. OR, odds ratio.; CI, confidence interval; Interim storage facility, ISF.

*p<0.05,

**p<0.001

## Discussion

Our study revealed differences in the levels of trust in information provided by government authorities regarding the ISF, with approximately 60% of study participants showing trust. These differences are due to factors such as gender, acceptance of the ISF, perceptions of genetic risks and negative effects, and mental health.

In particular, men demonstrated higher levels of trust in government-provided information compared to women. This was especially evident in Model II, focused on concerns about a negative image resulting from the nuclear accident. In contrast, for concerns specifically about genetic risks, gender does not appear to influence trust to the same degree. Freudenburg (1993) found similar gender differences in trust, noting these variations often depend on context; men typically show greater trust than women do when economic or societal aspects are emphasized [15]. Consistent with previous studies, our findings indicate that men showed significantly higher trust in information about the ISF than women, especially when concerns about radiation risks were not considered. This aligns with evidence suggesting that men generally express higher trust levels than women do and are more likely to consider the process of selecting storage sites to be fair and legitimate [14].

Furthermore, our data show that the perception of genetic risks due to radiation exposure decreases over time. In 2017, 72.5% of residents of Tomioka town expressed concerns about the genetic consequences of radiation exposure, whereas by 2021, this proportion had decreased significantly, to 48.1% [17]. According to data from 2022, the belief in the presence of genetic effects was highest among the group living outside Tomioka and Okuma (50.8%), followed by residents of Futaba town (31.5%), those living in Tomioka or Okuma (25.3%), and residents of Kawauchi village (24.4%). These studies indicate that people living somewhat farther from the immediate site of the accident and those who are unsure of their intention to return experience greater concerns about potential hereditary effects, possibly due to differences in access to information, risk perception, or the level of trust in official reports on radiation safety [16,17].

We observed a strong association between higher trust in the information provided by the public authorities and a greater acceptance of the ISF. Those who accept the presence of the storage facilities demonstrated greater trust in the information provided. These results were related to findings in the literature, which emphasize that political parties and responsible institutions view public trust as a key condition for a population's acceptance of nuclear infrastructure. Specifically, research has shown that trust is closely related to procedural acceptance, which, in turn, is linked to the decision-making process regarding the siting of facilities and, subsequently, to the acceptance of the repository itself [18]. Current discussions about building trust through management also highlight that trust plays an important role in reducing risk perception: the greater the level of trust, the lower the risk perception. In our study, "risk perception" refers to individuals' subjective judgments regarding the likelihood and severity of potential adverse consequences from radiation exposure and nuclear-related activities. We operationalized this concept through specific survey questions assessing respondents' concerns about genetic risks and the potential negative image of their community following the nuclear accident. Table 3 shows that in Model I, concerns about radiation-related genetic risks emerged as a strong negative predictor of trust (OR = 0.19), indicating that those worried about potential genetic effects of radiation are significantly less likely to trust official information. Model II, on the other hand, focused on concerns about a negative image resulting from the nuclear accident, which also reduced participants' trust (OR = 0.56), though the magnitude of this effect was not as large as that observed for genetic-risk concerns. This difference suggests that apprehensions about direct health risks (i.e., genetic effects) may have a more substantial impact on trust than broader concerns about stigma or image. Despite gender differences, several factors consistently showed similar effects across both models. Notably, participants who accept the ISF in Okuma and Futaba were significantly more likely to trust public authorities (ORs ~ 3.9–4.1), and those with good mental health showed increased trust (ORs ~ 1.75–2.01). This consistency highlights the overarching importance of supportive attitudes toward the ISF and mental well-being in shaping trust, regardless of the specific type of risk concern. These findings collectively suggest that while both dimensions of concern reduce trust, they do so to different extents, and their interaction with demographic factors (especially gender) varies. Understanding these nuanced differences can help guide more

targeted communication strategies—efforts that address health-related anxieties and stigma-related concerns separately may be more effective in maintaining or increasing public trust in nuclear-related facilities. While our analysis highlights an inverse association between trust and perceived risk, we recognize that risk perception is influenced by multiple factors beyond trust alone, such as familiarity, understanding, uncertainty, culture, and individual worldviews [19,20]. It is suggested that trust may influence risk perception in several ways, where perceived transparency and openness of participation can gain citizens' confidence that responsible institutions and agents are doing everything possible to find the safest solution [18]. The acceptance of nuclear facilities is formed based on the assessment of both their benefits and risks, in which trust plays a decisive role. Although our data cannot directly confirm causality, previous studies suggest that trust in the information provided by authorities typically precedes public acceptance of controversial or risk-related facilities [18].

Trust-building activities such as transparent communication, holding public forums, and independent monitoring (as exemplified by the Reprun Information Center launched in August 2018 by the Japanese Ministry of the Environment, where residents independently measure radiation levels and consult with experts) appear to have successfully established trust among residents. Such outreach efforts likely foster an environment conducive to acceptance, particularly when residents perceive these efforts to be credible, fair, and responsive to their concerns.

Several psychological and sociological factors may help explain why mental health can influence trust in government-provided information about the ISF. Good mental health often supports cognitive functions, such as information processing, critical thinking, and decision-making, potentially enabling the clearer and more objective evaluation of complex or emotionally charged information [21]. Stable mental health is also associated with emotional regulation and resilience helping individuals manage anxiety or stress related to sensitive topics like radiation waste management. Reduced anxiety can encourage a more balanced view of information and be less influenced by emotional reactions, which might otherwise lead to mistrust [22]. Importantly, however, experiencing poorer mental health does not inherently diminish an individual's capacity to critically assess or trust information; rather, it highlights the importance of employing tailored and empathetic communication strategies that directly address the emotional and psychological dimensions of risk perception. In our study, the percentage of women reporting poor mental health (44.8%) was slightly higher than men (39.8%). This finding may partially explain why women exhibited lower levels of trust in government-provided information, suggesting that mental health status could contribute to observed gender differences in trust. Nonetheless, other factors—such as gender-specific perceptions of risk, social expectations, and emotional or affective responses—likely play significant roles. Recognizing these complexities, public communication strategies should incorporate both emotional reassurance and cognitive clarity, acknowledging mental health as a crucial consideration in building public trust, especially among groups who might be experiencing greater psychological vulnerability or heightened anxiety regarding radiation-related issues [23]. Future research should explore further how gender, mental health, and other socio-cultural factors intersect to shape public trust in radiation-related contexts.

Our results confirm that trust in information about radiation risks is determined not only by the factual knowledge of the source but also by the public's perception of its impartiality and possible vested interests. Hunt and Frewer (1999) investigated public trust in the United Kingdom concerning various sources of information about radiation risks [24] and found that trust in sources depends on their perceived truthfulness (message bias) and expertise (degree of knowledge). Sources considered more truthful and knowledgeable elicited greater trust. For example, university scientists, the Department of Health, and local general practitioners ranked high in trust, due to low bias and a focus on public welfare. In contrast, government ministers and tabloid newspapers received low trust ratings because of high message bias and presumed vested interests. In the context of our study, Nagasaki University played a significant role in supporting education and advisory work with the population in the affected areas, including research programs aimed at risk communication, reducing public anxiety through objective risk assessment, and helping the public to understand radiation safety measures. This confirms the theory of trust in academia and institutions. Cooperation among governmental structures and independent scientific institutions strengthens the perception of expertise and impartiality—two factors critically important

for effective risk communication. Understanding that the perceptions of both knowledge and impartiality are extremely important, we observed that acceptance of the ISF is greater among people who believe they receive reliable and unbiased information. Effective communication about radiation risks is best carried out by independent organizations that possess technical knowledge and have demonstrated a clear commitment to the public interest [24]. In addition, recent research on radiation risk communication has shifted toward a heuristic approach, suggesting that the public's perception of risk is shaped substantially by affective or intuitive judgments, rather than by factual understanding alone [25,26]. For example, people often rely on intuitive shortcuts or emotional heuristics—such as feelings, social trust, and perceived benefits—to form judgments about radiation-related risks [27]. Thus, while providing factual information and demonstrating expertise remains crucial, effective risk communication should also address emotional, intuitive, and affective dimensions. Outreach efforts that foster positive feelings, empathy, and emotional reassurance can thus play a significant role in enhancing public trust, especially regarding complex and emotionally charged issues like nuclear facility siting.

## Conclusion

Our study highlights the crucial role of trust as a foundational factor influencing public acceptance of radiation-related facilities. Effective risk communication strategies should prioritize building trust as a first step, rather than assuming it will naturally follow acceptance. Achieving this requires authorities to not only demonstrate expertise and impartiality but also to directly and transparently address specific community concerns—particularly those related to genetic risks, environmental impacts, and negative perceptions linked to the nuclear accident. Collaborations with independent scientific institutions can further strengthen perceptions of transparency and credibility. Importantly, establishing trust is not merely about information dissemination; it demands active and ongoing community engagement informed by both quantitative assessments and qualitative insights into the community's values and concerns. Such strategies are essential for the successful implementation of radiation-related projects and for fostering public confidence in the authorities responsible for managing these initiatives.

## Limitations of the study

Our survey findings are specific to Tomioka, Futaba, and Okuma and may not apply elsewhere. Cultural and social factors can shape trust differently in other regions. Self-reported data may involve biases, and as a cross-sectional study it does not definitively establish the causal direction between trust and the acceptance of the ISF. Although our findings demonstrate a strong association between trust and acceptance, we cannot conclusively determine whether trust precedes acceptance or vice versa. Also, we did not examine factors such as media exposure, past government interactions, or radiation knowledge, which might influence trust. Our measurement of risk perception relied primarily on respondents' concerns about radiation-related genetic risks and their community's image. However, we did not explicitly measure dimensions such as their level of scientific understanding, uncertainty regarding risk estimates, or cultural and worldview factors. Future studies should incorporate these aspects to provide a more comprehensive understanding of risk perception and its interaction with trust including the measurement of heuristic predictors.

## Supporting information

**S1 Data. Questionnaire variables (categorical and continuous) were used in this study.**
(XLSX)

## Acknowledgments

We would like to thank all study participants and staff members in Okuma, Futaba, and Tomioka towns, Fukushima Prefecture, Japan.

 

## Author contributions

**Conceptualization:** Aizhan Zabirova, Hitomi Matsunaga.

**Data curation:** Yuya Kashiwazaki, Xu Xiao.

**Formal analysis:** Aizhan Zabirova, Hitomi Matsunaga.

**Visualization:** Hitomi Matsunaga.

**Writing – original draft:** Aizhan Zabirova.

**Writing – review & editing:** Thierry Schneider, Noboru Takamura.

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
