## [Decision Letter · Decision Letter 0]

27 Feb 2025

PONE-D-24-58280Understanding public trust in information about interim nuclear waste storage: The roles of acceptance, gender, and proximityPLOS ONE

Dear Dr. Matsunaga,

Thank you for submitting your manuscript to PLOS ONE. After careful consideration, we feel that it has merit but does not fully meet PLOS ONE’s publication criteria as it currently stands. Therefore, we invite you to submit a revised version of the manuscript that addresses the points raised during the review process.

We look forward to receiving your revised manuscript.

Kind regards,

Sakae Kinase, Ph.D.

Academic Editor

PLOS ONE

2. In the online submission form, you indicated that [We can share our data if somebody contacts us.].

Additional Editor Comments:

Your paper has been carefully considered by two referees. Your study is well organized and your paper is well written. However, there are some proposals for further improvements to make. Please carefully consider the two referees’ comments, it is recommended that you revise the paper accordingly.

Reviewers' comments:

Reviewer's Responses to Questions

**Comments to the Author**

1. Is the manuscript technically sound, and do the data support the conclusions?

Reviewer #1: Yes

Reviewer #2: Yes

2. Has the statistical analysis been performed appropriately and rigorously? 

Reviewer #1: Yes

Reviewer #2: Yes

3. Have the authors made all data underlying the findings in their manuscript fully available?

Reviewer #1: No

Reviewer #2: No

4. Is the manuscript presented in an intelligible fashion and written in standard English?

Reviewer #1: Yes

Reviewer #2: Yes

5. Review Comments to the Author

Reviewer #1: Thank you for your submission the manuscript is very good. I recommend minor revision based on following suggestions and information.

Statistical Methods, Lines 140 – 142 Suggest including some details about the differences between the logistic regression models. This detail is relevant to discussion about the regression results.

Discussion, Line 237 – Authors refer to their study from 2017. There does not appear to be a reference. Suggestion making this reference clear,

Discussion, Line 234 – “men expressed more greater in information”. The meaning is not clear. Suggest reviewing and revising.

Table 1, Note, Line 180 – “intention to return” appears twice. Suggest remove the duplication.

Introduction, Line 72-73 – BANANA = Build Absolutely Nothing Anywhere Near Anything. NOTE = Not Over There Either. For the information of the authors.

Reviewer #2: Comments are attached, as well as copied here.

Dear Editor,

I declare no conflict of interest in carrying out this review.

I will first summarize the research and provide my overall impression, then I will provide evidence and examples.

The authors have endeavored to investigate the relationship between public trust in public authorities responsible for nuclear waste storage by considering the impacts of acceptance, gender and proximity. A survey was conducted among 1,558 former residents of Tomioka, Okuma, and Futuba, who were 18 years or older at the time of the Fukushima Daiichi nuclear power plant accident. Given the time lapse since the accident, most survey respondents were generally older than 65 years. Residents from these towns have taken part in previous research studies employing surveys in the past. The survey aimed to determine participants' trust in the information provided by public authorities regarding the interim storage facility. Generally, the authors found that men and younger persons exhibited higher levels of trust, and overall, 60% of study participants showed trust. Two distinct logistic regression models were employed to prevent potential bias from a single variable dominating the regression analysis. Specifically, Model I considered concerns of genetic risk, while model II considered negative image concerns. Concerns about genetic risks and negative images significantly decrease trust in both models. Good mental health significantly increases trust in both models. The methodology undertaken by the authors follows earlier published research from their team. The results generally support the conclusions of the study, however, several points of clarification are raised.

Specific comments are provided below for the authors consideration.

Comment 1: There is a great deal of literature grounded in psychology on trust, and the factors that influence trust. It is unclear whether the survey included a definition of trust so that all respondents had the same interpretation. The literature (for a meta-analysis see doi: 10.3389/fpsyg.2023.1081086) explains predictive factors of trustworthiness as an outcome (rather than asking about trustworthiness as a direct question). Important predictive factors include reputation of the trustee and the closeness of relationship between trustor and trustee. A description should be added to the manuscript as to what is meant by “factors”, how those “factors” were chosen, and what supporting literature is aligned with the (somewhat limited) choice of “factors”. Other research indicates that trust can be broken down into several factors: humanity, capability, transparency, and reliability (or a variation of these). For this reason, survey questions (to evaluate trust) should be framed as “was the information reliable/transparent?”, “do you think the authority in question is capable/competent?”, etc.

Comment 2: For completeness, consider adding the survey as a stand-alone electronic attachment to the article.

Comment 3: In the Discussion, one could challenge that your “analysis confirms that trust in information about radioactive waste and its storage depends significantly on the public acceptance of facilities”. Rather, does public acceptance depend on trust? Thus, do the questions you are posing in the survey lead to the appropriate conclusions? Or should the conclusion be reframed to discuss how trust is first established (and discuss how this occurred through outreach and engagement activities), followed by acceptance.

Comment 4: The concept of risk perception is explored but ill-defined. Suggest defining risk perception more clearly, including how risk perception was measured. The relationship between a greater level of trust and a lower risk perception is vague and should be qualified. Is risk perception simply the individual’s belief that genetic risks due to radiation exposure may or may not occur and general concerns about negative perceptions? The way in which individual’s formulate their personal risk perception, or how they judge a risk, is influenced by many factors that are not discussed or accounted for, including familiarity with the risk, understanding of the risk, uncertainty surrounding the risk, not to mention culture and worldview, among other important factors that should be either added to the discussion or mentioned in the limitations, separate from factors that impact trust.

Comment 5: The reference of Hunt and Frewer (1999), while groundbreaking at the time, is now 25 years out of date. Do the conclusions (perceived truthfulness/message bias and expertise/degree of knowledge) hold true today? How do they compare to the current literature published by social scientists touching on ionizing radiation. For example, newer research is focused on reducing the importance on knowledge and focuses instead on heuristic predictors (e.g., DOI: 10.1097/HP.0b013e31823fb5a5).

Comment 6: Also in the Discussion, the tone could be framed more positively with regards to mental health. As written the Discussion left the impression that people who had less stable mental health struggled to trust information or accept a facility. The fundamental issue is not resilience of the community (or the individual), but whether the authority deserves trust (are they capable, transparent, reliable, do they demonstrate humanity). Individuals with less stable mental health can still make that determination if presented with information. Further, of the survey population, were the woman those with reported lower mental health? If so, please comment on how this variable may be impacting your conclusions.

Thank you for the invitation to peer-review the manuscript.

Please reach out directly to me should you have any additional questions, or comments, or points of clarification.

Kind Regards.

6. PLOS authors have the option to publish the peer review history of their article (what does this mean? ). If published, this will include your full peer review and any attached files.

**Do you want your identity to be public for this peer review?** For information about this choice, including consent withdrawal, please see our Privacy Policy .

Reviewer #1: No

Reviewer #2: No

---

## [Author Response · Author response to Decision Letter 1]

1 Apr 2025

Reviewer #1: Thank you for your submission the manuscript is very good. I recommend minor revision based on following suggestions and information.

Statistical Methods, Lines 140 – 142 Suggest including some details about the differences between the logistic regression models. This detail is relevant to discussion about the regression results.

Discussion, Line 237 – Authors refer to their study from 2017. There does not appear to be a reference. Suggestion making this reference clear,

Discussion, Line 234 – “men expressed more greater in information”. The meaning is not clear. Suggest reviewing and revising.

Table 1, Note, Line 180 – “intention to return” appears twice. Suggest remove the duplication.

Introduction, Line 72-73 – BANANA = Build Absolutely Nothing Anywhere Near Anything. NOTE = Not Over There Either. For the information of the authors.

Thank you for these recommendations. We have made minor revisions to the introduction, statistical methods, and discussion sections according to your suggestions. Please see:

p. 4, lines 72– 73

p. 8, lines 154–157

p. 14, lines 249– 251

p. 14, lines 255–257

Reviewer #2:

Comment 1: There is a great deal of literature grounded in psychology on trust, and the factors that influence trust. It is unclear whether the survey included a definition of trust so that all respondents had the same interpretation. The literature (for a meta-analysis see doi: 10.3389/fpsyg.2023.1081086) explains predictive factors of trustworthiness as an outcome (rather than asking about trustworthiness as a direct question). Important predictive factors include reputation of the trustee and the closeness of relationship between trustor and trustee. A description should be added to the manuscript as to what is meant by “factors”, how those “factors” were chosen, and what supporting literature is aligned with the (somewhat limited) choice of “factors”. Other research indicates that trust can be broken down into several factors: humanity, capability, transparency, and reliability (or a variation of these). For this reason, survey questions (to evaluate trust) should be framed as “was the information reliable/transparent?”, “do you think the authority in question is capable/competent?”, etc.

Thank you for your insightful comments regarding the conceptualization and measurement of trust. We appreciate the opportunity to clarify how “trust” was defined and operationalized in our study. In our questionnaire, we asked a general question (“Do you trust the information provided by government organizations about temporary storage facilities?”) without providing a specific, standardized definition of trust. We understand that trust is a multifaceted concept, including aspects such as reliability, transparency, and the competence of the source. However, in this study, the aim was to measure trust in information as such, which is in line with the approaches of a number of previous studies (e.g., Fukasawa et al., 2020; Guizhen He et al., 2014; Hobson, 2015).

In response to your recommendation to clarify the question of “factors,” we have added information in the introduction describing the criteria we used to select variables that potentially influence trust; please see p. 5, lines 91–106.

Comment 2: For completeness, consider adding the survey as a stand-alone electronic attachment to the article.

Thank you for this comment. The survey has now been added as an appendix.

Comment 3: In the Discussion, one could challenge that your “analysis confirms that trust in information about radioactive waste and its storage depends significantly on the public acceptance of facilities”. Rather, does public acceptance depend on trust? Thus, do the questions you are posing in the survey lead to the appropriate conclusions? Or should the conclusion be reframed to discuss how trust is first established (and discuss how this occurred through outreach and engagement activities), followed by acceptance.

Thank you for raising this point. We agree that our original wording might suggest a causal direction that requires clarification. Indeed, our analysis was cross-sectional, thus causality cannot be inferred directly. While we initially framed our conclusion as "trust depending on acceptance," we recognize the possibility of reverse causality—existing trust levels may actually influence the acceptance of information provided by public authorities. Given our survey’s cross-sectional nature, we cannot definitively establish whether trust precedes acceptance or vice versa; however, our logistic regression analyses clearly show a strong association between acceptance of the interim storage facilities (ISF) and trust in information provided by authorities. Although our current data cannot determine temporal order or causality, the existing literature suggests that trust is typically a prerequisite for acceptance in contexts involving high perceptions of risks, such as radioactive waste management (e.g., Slovic et al., 1991; Renn, 2008).

With the above in mind, we concur with your suggestion to reframe our conclusion. Rather than implying trust as a result of acceptance, we will emphasize the importance of first building trust through effective outreach, transparent communication, and meaningful engagement activities as essential steps toward fostering acceptance. Future research, ideally using longitudinal data, would help to clarify the precise causal pathways between trust establishment and public acceptance of nuclear-related facilities.

We have revised the discussion accordingly to reflect this nuance:

p. 14, lines 264–265

p. 16, lines 302–304

p. 19, lines 363–374

p.19, lines 379–388

Comment 4: The concept of risk perception is explored but ill-defined. Suggest defining risk perception more clearly, including how risk perception was measured. The relationship between a greater level of trust and a lower risk perception is vague and should be qualified. Is risk perception simply the individual’s belief that genetic risks due to radiation exposure may or may not occur and general concerns about negative perceptions? The way in which individual’s formulate their personal risk perception, or how they judge a risk, is influenced by many factors that are not discussed or accounted for, including familiarity with the risk, understanding of the risk, uncertainty surrounding the risk, not to mention culture and worldview, among other important factors that should be either added to the discussion or mentioned in the limitations, separate from factors that impact trust.

Thank you for highlighting this important point. We agree that our manuscript would benefit from a clearer definition of risk perception, including explicit details on its measurement and the factors involved. In response, we have revised the manuscript to clarify the definition and operationalization of risk perception. Specifically, we have now explicitly defined risk perception as "an individual's subjective evaluation of the likelihood and severity of potential adverse consequences of radiation exposure and associated nuclear facilities" (adapted from Slovic, 1987). In our study, risk perception was assessed operationally through survey items asking respondents about their concerns regarding radiation-related genetic risks and concerns about a negative image arising from the nuclear accident. These two variables served as proxies for participants' subjective perceptions of radiation-related risks. Additionally, we have clarified in the discussion that the observed relationship between trust and risk perception is consistent with the existing literature, indicating that increased trust typically lowers perceived risk (Renn & Levine, 1991; Slovic, 1993). At the same time, we acknowledge that risk perception is a multifaceted construct influenced by several factors beyond trust, such as familiarity with the hazard, understanding of the risk, perceived uncertainty, and cultural beliefs and values. Given our study’s design, we could not fully explore all of these dimensions comprehensively, so we have clarified this limitation in the manuscript, noting that future research should incorporate these additional factors to enhance understanding of how risk perception develops independently and interacts with trust.

We have revised the discussion section and limitations accordingly, to reflect these clarifications and the reviewer’s valuable suggestions; please see pp. 15–16, lines 273– 297.

Comment 5: The reference of Hunt and Frewer (1999), while groundbreaking at the time, is now 25 years out of date. Do the conclusions (perceived truthfulness/message bias and expertise/degree of knowledge) hold true today? How do they compare to the current literature published by social scientists touching on ionizing radiation. For example, newer research is focused on reducing the importance on knowledge and focuses instead on heuristic predictors (e.g., DOI: 10.1097/HP.0b013e31823fb5a5).

Thank you for this meaningful comment. You are correct, the original reference (Hunt and Frewer, 1999) reflects a somewhat earlier understanding of the determinants of trust, focusing significantly on the perceived knowledge and impartiality of information sources. However, as you have pointed out, more recent literature has shifted toward understanding the role of heuristic predictors—such as affective responses, social trust, perceived benefits, and intuitive evaluations—in shaping public perceptions of risks related to ionizing radiation (e.g., Slovic, 2012; Perko, 2014; Peters et al., 2012, DOI: 10.1097/HP.0b013e31823fb5a5). Accordingly, we have revised our discussion to incorporate newer perspectives that emphasize heuristics. Specifically, we have clarified that, while perceived expertise and impartiality remain relevant, heuristic factors such as affective judgments, intuitive risk evaluations, and social trust significantly influence trust formation. This newer perspective emphasizes the importance of addressing the intuitive and emotional dimensions of risk perception, rather than solely improving the public's knowledge or cognitive understanding. We have updated our discussion and added relevant citations from contemporary studies on heuristic predictors and affective dimensions of risk perception associated with ionizing radiation. The study limitations have also been updated, to acknowledge that our research did not explicitly measure heuristic predictors and recommending this as an area for future exploration.

We have incorporated these changes into the manuscript, updating our references and explicitly noting the need to explore heuristic predictors more thoroughly in future research; please see p. 18, lines 352– 361.

Comment 6: Also in the Discussion, the tone could be framed more positively with regards to mental health. As written the Discussion left the impression that people who had less stable mental health struggled to trust information or accept a facility. The fundamental issue is not resilience of the community (or the individual), but whether the authority deserves trust (are they capable, transparent, reliable, do they demonstrate humanity). Individuals with less stable mental health can still make that determination if presented with information. Further, of the survey population, were the woman those with reported lower mental health? If so, please comment on how this variable may be impacting your conclusions.

Thank you for highlighting the importance of positively and sensitively framing the discussion on mental health. We fully agree that mental health should not be portrayed as a limitation of individuals' capacities to trust, but rather as an important factor influencing how risk information is processed emotionally. In response to your valuable comment, we have revised our discussion to emphasize that individuals experiencing poorer mental health may benefit from tailored, empathetic, and emotionally supportive communication approaches.

Additionally, we analyzed our data further regarding gender differences and mental health. We found that, although men were more numerous overall among the study’s participants, women reported a slightly higher percentage of poorer mental health (44.8%), compared to men (39.8%). This finding could partially explain the observed lower trust levels among women, although other factors (such as gender-specific risk perceptions and socio-cultural influences) are also likely influential.

We have included these clarifications in the revised discussion and highlighted the complexity of these relationships, recommending further targeted research in the future; please see p. 17, lines 318 –331.

---

## [Decision Letter · Decision Letter 1]

16 Apr 2025

Understanding public trust in information about interim nuclear waste storage: The roles of acceptance, gender, and proximity

PONE-D-24-58280R1

Dear Dr. Matsunaga,

We’re pleased to inform you that your manuscript has been judged scientifically suitable for publication and will be formally accepted for publication once it meets all outstanding technical requirements.

Kind regards,

Sakae Kinase, Ph.D.

Academic Editor

PLOS ONE

Additional Editor Comments (optional):

I have much pleasure in recommending this paper for publication. The manuscript has been substantially with changes according to reviewers' comments.

Reviewers' comments:

Reviewer's Responses to Questions

**Comments to the Author**

1. If the authors have adequately addressed your comments raised in a previous round of review and you feel that this manuscript is now acceptable for publication, you may indicate that here to bypass the “Comments to the Author” section, enter your conflict of interest statement in the “Confidential to Editor” section, and submit your "Accept" recommendation.

Reviewer #1: All comments have been addressed

Reviewer #2: All comments have been addressed

2. Is the manuscript technically sound, and do the data support the conclusions?

Reviewer #1: Yes

Reviewer #2: Yes

3. Has the statistical analysis been performed appropriately and rigorously? 

Reviewer #1: Yes

Reviewer #2: Yes

4. Have the authors made all data underlying the findings in their manuscript fully available?

Reviewer #1: Yes

Reviewer #2: Yes

5. Is the manuscript presented in an intelligible fashion and written in standard English?

Reviewer #1: Yes

Reviewer #2: Yes

6. Review Comments to the Author

Reviewer #1: (No Response)

Reviewer #2: Thank you for addressing all of my comments, the manuscript reads well, and presents a more objective stance now.

7. PLOS authors have the option to publish the peer review history of their article (what does this mean? ). If published, this will include your full peer review and any attached files.

**Do you want your identity to be public for this peer review?** For information about this choice, including consent withdrawal, please see our Privacy Policy .

Reviewer #1: No

Reviewer #2: No

---

## [Editor Report · Acceptance letter]

PONE-D-24-58280R1

PLOS ONE

Dear Dr. Matsunaga,

I'm pleased to inform you that your manuscript has been deemed suitable for publication in PLOS ONE. Congratulations! Your manuscript is now being handed over to our production team.

Kind regards,

on behalf of

Professor Sakae Kinase

Academic Editor

PLOS ONE